# Activation of targetable inflammatory immune signaling is seen in myelodysplastic syndromes with SF3B1 mutations

Gaurav S Choudhary[1†], Andrea Pellagatti[2†], Bogos Agianian[3], Molly A Smith[4], Tushar D Bhagat[1], Shanisha Gordon-Mitchell[1], Srabani Sahu[1], Sanjay Pandey[1], Nishi Shah[1], Srinivas Aluri[1], Ritesh Aggarwal[1], Sarah Aminov[1], Leya Schwartz[1], Violetta Steeples[2], Robert N Booher[5], Murali Ramachandra[6], Maria Samson[5], Milagros Carbajal[1], Kith Pradhan[1], Teresa V Bowman[1], Manoj M Pillai[7], Britta Will[1], Amittha Wickrema[8], Aditi Shastri[1], Robert K Bradley[9], Robert E Martell[5], Ulrich G Steidl[1], Evripidis Gavathiotis[3], Jacqueline Boultwood[2]*, Daniel T Starczynowski[4]*, Amit Verma[1]*

[1]Blood Cancer Institute, Albert Einstein College of Medicine, Montefiore Medical Center, The Bronx, United States; [2]Nuffield Division of Clinical Laboratory Sciences, Radcliffe Department of Medicine, University of Oxford, Oxford, United Kingdom; [3]Department of Biochemistry, Albert Einstein College of Medicine, The Bronx, United States; [4]Experimental Hematology and Cancer Biology, Cincinnati Children's Hospital Medical Center, Cincinnati, United States; [5]Curis Inc, Lexington, United States; [6]Aurigene Inc, Bangalore, India; [7]Yale University, New Haven, United States; [8]University of Chicago, Chicago, United States; [9]Fred Hutchinson Cancer Center, Seattle, United States

*For correspondence:
jacqueline.boultwood@ndcls.ox.
ac.uk (JB);
daniel.starczynowski@cchmc.org
(DTS);
amit.verma@einsteinmed.edu
(AV)

†These authors contributed
equally to this work

Competing interest: See page
15

Reviewing Editor: Jameel Iqbal,
DaVita Labs, United States

## Abstract

**Background:** Mutations in the *SF3B1* splicing factor are commonly seen in myelodysplastic syndromes (MDS) and acute myeloid leukemia (AML), yet the specific oncogenic pathways activated by mis-splicing have not been fully elucidated. Inflammatory immune pathways have been shown to play roles in the pathogenesis of MDS, though the exact mechanisms of their activation in splicing mutant cases are not well understood.

**Methods:** RNA-seq data from *SF3B1* mutant samples was analyzed and functional roles of interleukin-1 receptor-associated kinase 4 (*IRAK4*) isoforms were determined. Efficacy of *IRAK4* inhibition was evaluated in preclinical models of MDS/AML.

**Results:** RNA-seq splicing analysis of *SF3B1* mutant MDS samples revealed retention of full-length exon 6 of *IRAK4*, a critical downstream mediator that links the Myddosome to inflammatory NF-kB activation. Exon 6 retention leads to a longer isoform, encoding a protein (IRAK4-long) that contains the entire death domain and kinase domain, leading to maximal activation of NF-kB. Cells with wild-type *SF3B1* contain smaller IRAK4 isoforms that are targeted for proteasomal degradation. Expression of IRAK4-long in *SF3B1* mutant cells induces TRAF6 activation leading to K63-linked ubiquitination of CDK2, associated with a block in hematopoietic differentiation. Inhibition of IRAK4 with CA-4948, leads to reduction in NF-kB activation, inflammatory cytokine production, enhanced myeloid differentiation in vitro and reduced leukemic growth in xenograft models.

**Conclusions:** *SF3B1* mutation leads to expression of a therapeutically targetable, longer, oncogenic *IRAK4* isoform in AML/MDS models.

**Funding:** This work was supported by Cincinnati Children's Hospital Research Foundation, Leukemia Lymphoma Society, and National Institute of Health (R35HL135787, RO1HL111103, RO1DK102759, RO1HL114582), Gabrielle's Angel Foundation for Cancer Research, and Edward P. Evans Foundation grants to DTS. AV is supported by Edward P. Evans Foundation, National Institute of Health (R01HL150832, R01HL139487, R01CA275007), Leukemia and Lymphoma Society, Curis and a gift from the Jane and Myles P. Dempsey family. AP and JB are supported by Blood Cancer UK (grants 13042 and 19004). GC is supported by a training grant from NYSTEM. We acknowledge support of this research from The Einstein Training Program in Stem Cell Research from the Empire State Stem Cell Fund through New York State Department of Health Contract C34874GG. MS is supported by a National Institute of Health Research Training and Career Development Grant (F31HL132420).

## Editor's evaluation

This is an outstanding manuscript that sought to evaluate a mutation commonly seen in AML and MDS in the splicesome SF3B1. The authors demonstrate that this mutation leads to a shift in the production of a long-form of IRAK4 (called IRAK4-L), which is part of inflammatory signaling in immune cells. They demonstrate that IRAK4-L stabilizes the cell cycle protein CDK2, and targeting IRAK4-L with an inhibitor can induce differentiation and slow clonal uptake in murine transplantation models.

## Introduction

Myeloid malignancies myelodysplastic syndromes (MDS) and acute myeloid leukemia (AML) are associated with dismal prognosis and need newer therapeutic options. Mutations in spliceosome genes are commonly seen in MDS and AML and *SF3B1* is the most frequently mutated gene in patients with MDS (*Yoshida et al., 2011*). Even though reports have shown that mutations in *SF3B1* can lead to widespread changes in splicing (*Inoue and Abdel-Wahab, 2016*), the exact pathways that are disrupted and lead to the pathogenesis of ineffective hematopoiesis and carcinogenesis in MDS and AML are not yet fully elucidated.

The innate immune signaling pathways have been shown to be activated in MDS and AML (*Barreyro et al., 2018*). The upstream activation of toll like receptors (TLRs), interleukin receptor 1 receptor accessory protein (IL1RAP), and interleukin 8 (IL8) activate the IRAK/TRAF6 pathways and lead to NF-kB activation (*Mitchell et al., 2018*). Even though these pathways have been shown to be overactivated, a link between this activation and commonly seen *SF3B1* mutation has not been established. We had recently shown that another splicing mutation in *U2AF1* splicing factor can lead to retention of exon 4 leading to overexpression of *IRAK4* in MDS and AML (*Smith et al., 2019*). In the present report, we now demonstrate that *SF3B1* mutation leads to exon 6 retention leading to overexpression of active isoform of IRAK4 known as IRAK4-long (IRAK4-L). We demonstrate that this IRAK4-L isoform containing the death domain can associate with MYD88 and lead to activation of downstream NF-kB signaling. These data demonstrate the link between *SF3B1* mutation and oncogenic IRAK4 signaling in MDS/AML and demonstrate the therapeutic potential of inhibiting this pathway via pharmacological inhibitors.

## Materials and methods

(Study schema outlined in *Figure 5—figure supplement 5*).

### Cell lines and human samples

HEK-293T and THP1 were purchased from the American Type Culture Collection. MDS-L was provided by Dr. Kaoru Tohyama (Kawasaki Medical School, Okayama, Japan) (*Matsuoka et al., 2010*; *Rhyasen et al., 2014*). Cell lines were cultured according to manufacturer's instruction in either RPMI 1640,

**eLife digest** Genes contain blocks of code that tell cells how to make each part of a protein. Between these blocks are sections of linking DNA, which cells remove when they are preparing to use their genes. Scientists call this process 'splicing'. Cells can splice some genes in more than one way, allowing them to make different proteins from the same genetic code.

Mutations that affect the splicing process can change the way cells make their proteins, leading to disease. For example, the myelodysplastic syndromes are a group of blood cancers often caused by mutations in splicing proteins, such as SF3B1. The disorder stops blood cells from maturing and causes abnormal inflammation. So far, the link between splicing, blood cell immaturity, inflammation and cancer is not clear.

To find out more, Choudhary, Pellagatti et al. looked at the spliced genetic code from people with myelodysplastic syndromes. Mutations in the splicing protein SF3B1 changed the way cells spliced an important signalling molecule known as IRAK4. Affected cells cut out less genetic code and made a longer version of this signalling protein, named IRAK4-Long. This altered protein activated inflammation and stopped blood cells from maturing. Blocking IRAK4-Long reversed the effects. It also reduced tumour formation in mice carrying affected human cells.

The molecule used to block IRAK4, CA-4948 – also known as Emavusertib – is currently being evaluated in clinical trials for myelodysplastic syndromes and other types of blood cancer. The work of Choudhary, Pellagatti et al. could help scientists to design genetic tests to predict which patients might benefit from this treatment.

IMDM, or DMEM medium supplemented with 10% FBS. MDS-L cell line was cultured in the presence of 10 ng/mL human recombinant IL-3 (Stem Cell Technologies). Cell lines were tested for mycoplasma and authenticated by STR. The MDS/AML patient samples used in this study were obtained with written informed consent under approval by the IRBs of the Albert Einstein College of Medicine. K562 isogenic cells with wild type (WT) and K700E mutation of SF3B1 were purchased from Horizon Discovery.

## Plasmids and reagents

The IRAK4 inhibitor CA-4948 was received from Curis Inc Dinaciclib and AT7519 was purchased from Selleck Chemicals. Recombinant human IL-1β and IL-3 was purchased from PeproTech. TLR-5 was purchased from Invivogen. *IRAK4*-S1 and *IRAK4*-S2 constructs were obtained from Integrated DNA Technologies IDT and cloned into pCDNA3.1 plasmid. *IRAK4*-L-Flag, pCNDA3.1, *MYD-88*-HA, *Traf6*-Flag was provided by Dr. Daniel Starczynowski (Cincinnati Children's Hospital Medical Center, Cincinnati). K48-HA, K63-HA, *CDK2*-Myc was purchased from Addgene. The plasmids were transfected using Lipofectamine 2000 Transfection Reagent (Thermo Fisher Scientific) according to manufacturer's instructions. CDK2 mutants were generated using QuikChange II XL Site-Directed Mutagenesis Kit (Agilent Technologies) siControl and siIRAK4 were purchased from Horizon Discovery and transfected into human AML samples using Amaxa Nucleofector Kit T (Lonza) (program number G-016) according to manufacturer's protocol.

## RNA sequencing of MDS CD34+ cells

Total RNA from MDS and control bone marrow CD34+ cells was extracted using TRIzol with a linear acrylamide carrier, treated with DNase I (Life Technologies) and purified using Agencourt RNAClean XP beads (Beckman Coulter). RNA quality was determined using an RNA 6000 Bioanalyzer pico kit (Agilent). A cDNA library was produced using a SMARTer library preparation protocol (Clontech). Sequencing (100 bp paired-end reads) was performed on an Illumina HiSeq4000. The reads were mapped to human genome GRCh37 using HISAT2 version 2.0.0-beta. Uniquely mapped read pairs

were counted using htseq-count. The data have been deposited in the NCBI's Gene Expression Omnibus (GEO) repository (accession number: GSE114922).

## Immunoblotting and immunoprecipitation

Total protein lysates were obtained from cells by lysing the samples in 1% NP-40 lysis buffer 20 mmol/l Tris-HCl, pH 7.5; 1 mmol/l EDTA; 150 mmol/l NaCl; (1% NP-40), containing phosphatase inhibitor cocktails 2 and 3 and protease inhibitors (Sigma-Aldrich), for 30–45 min at 4°C. Equal amount of protein was prepared by calculating protein concentration using Bradford reagent (Bio-Rad) and 50 µg of protein was resolved on 10–12% SDS-PAGE followed by transferring to nitrocellulose or PVDF membranes (EMD-Millipore). For immunoprecipitation, cells were lysed in 2% CHAPS lysis buffer (20 mmol/l Tris-HCl, pH 7.5; 150 mmol/l NaCl; 1 mmol/l EDTA; 2% CHAPS; Sigma-Aldrich) with protease and phosphatase inhibitors (Sigma-Aldrich) for 1 hr on ice. Protein lysates were incubated with primary antibody overnight at 4°C. Protein A agarose beads (Rockland Immunochemicals, Gilbertsville, PA) were added equally to all the samples and incubated for 1 hr 4°C. The beads were washed three times with CHAPS, eluted with loading buffer supplemented with 2-mercaptoethanol (Sigma-Aldrich) and western blotting was performed. In ubiquitination experiments, MG132 and N-ethylmaleimide (Sigma-Aldrich) were added to lysis buffer with dithiothreitol (Sigma-Aldrich) to prepare lysates. Western blot analysis was performed with the following antibodies: IRAK4 C-term (ab5985; Abcam), IRAK N-term (4363; Cell Signaling Technology), Flag (F3165; Sigma), IRAK1 (sc-7883; Santa Cruz), phospho-IRAK1 (T209) (A1074; AssaybioTech), IKKβ (2370; Cell Signaling Technology), phospho-IKKα/β (2697; Cell Signaling Technology), p38 MAPK (9212; Cell Signaling Technology), phopho-p38 MAPK (4631; Cell Signaling Technology), ERK (4695; Cell Signaling Technology), phospho-ERK (4377; Cell Signaling Technology), p-JNK (4668; Cell Signaling), p65 (8242; Cell Signaling Technology), phospho-p65 (3033; Cell Signaling Technology) and b-actin (Santa-Cruz Biotechnology).

## Flow cytometry

The antibodies used for flow cytometric analysis of human MDS and AML cells included Mouse CD45 FITC; Human CD45 PeCy7 (e-Bioscience); Human CD4-Pacific Orange; Human CD8-Pacific Orange; Human CD19-Pacific Blue; Human CD20-Pacific Blue; Human CD11b-APC (Thermo Fisher); Human CD34-PE (Miltenyi Biotec);and Human CD33-APC-Cy7 (Abcam).

## Clonogenic progenitor assays

For clonogenic assays with primary MDS/AML cells, primary patient samples and healthy controls (HC) were plated in methylcellulose (Stem Cell Technologies H4435, Vancouver, CA) with the CA-4948 and control, and colonies were counted after 14–17 days.

## Xenografts

This study was performed in strict accordance with the recommendations in the Guide for the Care and Use of Laboratory Animals of the National Institutes of Health. All the animals were handled according to approved institutional animal care and use committee (IACUC) protocols (001371) of the Albert Einstein College of Medicine. Mice were sacrificed and assessed for tumor burden measurements. For primary patient-derived xenografts NOD/SCID IL2Rgamma KO NSG mice were irradiated (200 rads) 24 hr prior to injection. Mononuclear cells from primary MDS patients were isolated by Ficoll separation. 2–5 × 10$^6$ MNCs were administered via tail vein injection. 3–4 weeks later, BM aspiration were performed and analyzed by flow cytometry for the human cell engraftment. Mice were considered to be engrafted if they showed 0.1% or higher human derived CD45+ cells. The engrafted mice were randomized for treatment with 12.5 mg/kg/d of CA-4948 or control 5 times a week for indicated times. Following treatment, BM aspirations and flow cytometry analysis for the above-mentioned markers were performed. All mice were bred, housed, and handled in the animal facility of Albert Einstein College of Medicine.

### Reporter cell lines

IRAK4 exon 6 and its flanking intronic sequences were inserted into the EcoRI and BamHI sites of pFlareA vector. pFlareA-exon4 vectors were linearized by DraIII and transfected into 293T cells. Cell colonies stably expressing the reporter were selected by 1 mg/mL G418 and stable expression of GFP and RFP.

### Polymerase chain reaction

Primer3 designed primers:

> Fw - TGAACGACCCATTTCTGTTGG
> Rev - GAGTCTGTCTAGCAATGAACCA
> Expected product sizes:
> Exon 6 short isoform – 194 bp
> Exon 6 long isoform – 280 bp

cDNA was generated using a High-Capacity cDNA Reverse Transcription Kit (Thermo Fisher). The primers listed above were used with Maxima Hotstart PCR mastermix to amplify the region containing exon 6 of IRAK4 and PCR products separated by electrophoresis on a 2% agarose gel.

### Peptide docking

3D structure of CDK2 peptide was created using BioLuminate (Schrodinger LLC, New York). The peptide was docked to TRAF6 substate binding domain (PDB code: 1LB5) using Glide (Schrodinger LLC) using 'SP Peptide' mode that ensures enhanced conformational sampling of flexible polypeptides (*Tubert-Brohman et al., 2013*). To assure more accurate pose ranking, docked peptide poses were scored using implicit solvent MM-GBSA calculations. Final MM-GBSA interaction energies were used to create and rank peptide pose clusters.

## Results

### *SF3B1* mutations are associated with inclusion of exon 6 in MDS

We sought to determine whether primary MDS samples with *SF3B1* mutations have altered splicing of transcripts involved in immune and inflammatory pathways. Analysis of RNA-seq of purified CD34+ HSPCs from HC and *SF3B1* mutant MDS samples demonstrated an increased retention of full length exon 6 of *IRAK4* in the *SF3B1* mutant samples (*Figure 1A*). The full length retention of exon 6 was confirmed by RT-PCR in a larger cohort of primary samples demonstrating a significantly greater ratio of long isoform of *IRAK4* in *SF3B1* mutant samples when compared to HC and MDS patients without splicing mutations (*Figure 1B and C*). To gain further insight into the regulation of *IRAK4* exon 6 inclusion in the presence of *SF3B1* mutations, we generated an *IRAK4* exon 6 cassette splicing reporter in HEK293 cells (HEK293-*IRAK4*exon6). Transfection of mutant *SF3B1*-K700E into HEK293-*IRAK4*exon6 cells resulted in inclusion of *IRAK4* exon 6 from the splicing reporter compared to *SF3B1*-WT or vector-transfected HEK293-*IRAK4*exon6 cells (*Figure 1D*), indicating that *SF3B1*-K700E directly mediates inclusion of *IRAK4* exon 6.

### *SF3B1* mutation leads to longer, stable, and functionally active isoform of IRAK4

To evaluate the effects of retention of full length exon 6 seen in *SF3B1* mutant samples, we modeled the exon usage in vitro. Full length IRAK4 protein (IRAK4-L) is composed of a death domain, hinge region, and a kinase domain (*Figure 2A*). Exon 6 encodes a part of the kinase domain of IRAK4 protein. The exclusion of mRNA sequence encoding amino acids 188–217 in exon 6 as seen in HC can lead to two possible smaller IRAK4 isoforms. Isoform S1 (191 amino acids) could be produced due to formation of a premature stop codon (TGA) leading to a smaller IRAK4 isoform containing

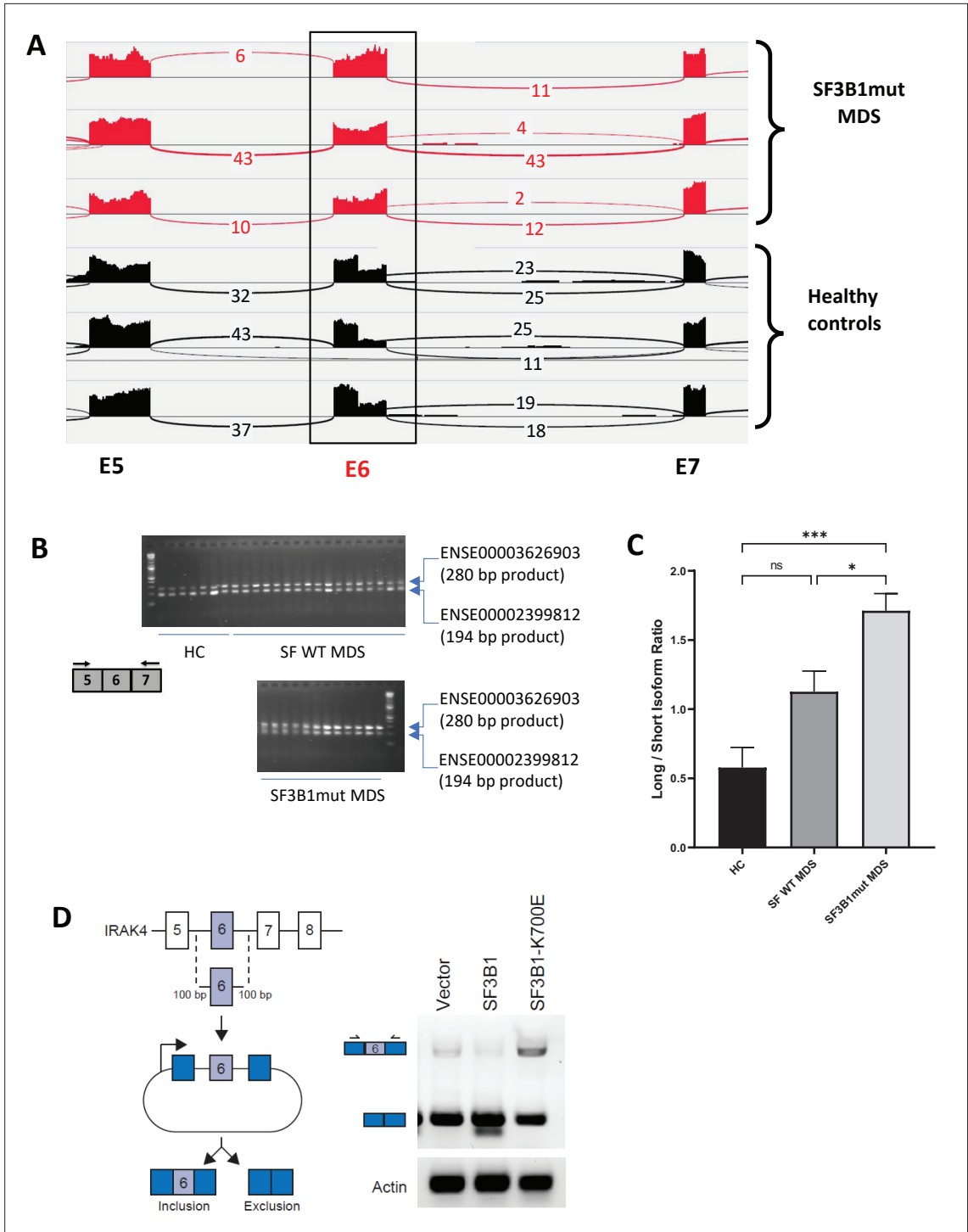

**Figure 1.** *SF3B1* mutations are associated with inclusion of exon 6 in myelodysplastic syndromes (MDS) . (**A**) Sashimi plots representing full length *IRAK4* exon 6 inclusion or exclusion in CD34+ cells from healthy controls (HC) (n=3), and from MDS patients with mutation in *SF3B1* (MDS-SF3B1[mut]; n=3) based on RNA-sequencing junction reads. (**B**) RT-PCR analysis of CD34+ cells from HC (n=7), from MDS patients with no splicing factor mutations (MDS-SF[WT]; n=17), and from MDS patients with mutation in *SF3B1* (MDS-*SF3B1*[mut]; n=12) using primers flanking IRAK4 exon 6. (**C**) Densitometric quantification of

*Figure 1 continued on next page*

Figure 1 continued

IRAK4 exon 6 inclusion calculated as the ratio of the long isoform versus the short isoform from panel B. Data represent the mean ± SEM. P-values were obtained using Kruskal–Wallis test with Dunn's multiple comparisons test. * p<0.05, *** p<0.001 (**D**) Schematic of *IRAK4* exon 6 splicing reporter. *IRAK4* exon 6 and 50 bp of flanking introns were cloned into a splicing reporter (pFlare5A)55 and stably expressed into HEK293 cells. *SF3B1* and *SF3B1-K700E* were transfected and IRAK4 exon 6 splicing was measured by RT-PCR using primers within pFlare5A. WT: wild type.

death, hinge, and part of kinase domain. Isoform S2 (243 amino acids) (S2) could also be generated by utilizing an alternative translational start site (ATG) resulting in an IRAK4 isoform with only the kinase domain (**Figure 2A**, **Figure 2—figure supplement 1**). We generated flag tagged plasmids of IRAK4 with full length exon 6 (IRAK4-L), and shorter S1 and S2 isoforms and transfected them in HEK-293T cells and evaluated their effects on NF-kB pathway activation. Immunoblotting analysis showed that plasmid with the full length exon 6 led to stable expression of IRAK4-long protein and resulted into increased phosphorylation of p65 when compared to the smaller isoforms (**Figure 2B**). By contrast, transfection of IRAK4 S1 and S2 constructs led to generation of lower molecular weight IRAK4 proteins with decreased expression when compared to the longer isoform (**Figure 2B**). Treatment with proteasome inhibitor (MG132) led to increased expression of the smaller IRAK4 bands demonstrating that these were targeted for proteasomal degradation (**Figure 2C**). Immunoprecipitation in HEK293T cells transfected with IRAK4-L, S1, S2, and HA-tagged Lys48 linked in presence of proteasome inhibitor MG132 demonstrated an increased association of Lys48 linked ubiquitin chains with shorter IRAK4 isoforms, validating ubiquitin mediated degradation (**Figure 2D**).

Next, we wanted to determine whether expression of the *SF3B1* mutation leads to changes in isoforms and downstream NF-kB activation. Isogenic cells with WT and K700E mutation of SF3B1 were evaluated and the mutant cells demonstrated overexpression of IRAK4-L and higher activation of p-p65 (**Figure 2E**). Finally, we showed that the IRAK4-long containing the death domain was able to associate with MYD88 in immunoprecipitation experiments, while the shorter isoforms did not; thus, providing the mechanistic basis of activation seen only with the IRAK4-long isoform (**Figure 2F**).

## CDK2 is ubiquitinated at K63 residues downstream of IRAK4/TRAF6 in *SF3B1* mutants

Having demonstrated overexpression of the active IRAK4-long isoform in *SF3B1* mutant samples, we next wanted to determine the downstream pathways that are activated by these changes. Signal transduction via TLRs is primarily mediated by IRAK4-TRAF6 activation and TRAF6 is an E3 ligase that conjugates ubiquitin chains to itself and its substrates. The consensus sequence of TRAF6 binding motif is P(P-2)XE(P0)XX(Acidic/Aromatic)(*P*+3) and is present in known TRAF6 substrates (**Figure 3A**). Since MDS/AML are characterized by ineffective hematopoiesis due to block in myeloid differentiation, we performed a search for the TRAF6 motif in cell cycle regulators and determined that CDK2 could be a potential substrate (**Figure 3A**). Structural modeling of TRAF6 and CDK2 peptide show that positions $P_{-2}$, $P_0$, and $P_3$ in CDK2 correspond to three distinct sub-pockets in the binding site of TRAF6 and supports the notion that CDK2 can be ubiquitinated by TRAF6 (**Figure 3B** and **Figure 3—figure supplement 1**). To directly test the role of TRAF6 dependent ubiquitination of CDK2 in chronic innate immune signaling, we ectopically expressed CDK2, IRAK4, and TRAF6 in presence of either Lys48 linked ubiquitin or Lys63 linked ubiquitin. CDK2 was associated with a much stronger K63 ubiquitin that was dependent on TRAF6 and IRAK4 overexpression (**Figure 3C**). To validate this result, we generated alanine and glycine mutants (CDK2 Myc 122 and CDK2 Myc 123) within CDK2 binding site of TRAF6 and examined their ubiquitination in presence of IRAK4-L and TRAF6 by immunoprecipitating CDK2. Immunoblotting analysis show that TRAF6 mediated Lys63 linked ubiquitination was decreased in CDK2 mutants as compared to wild type CDK2 (**Figure 3D**).

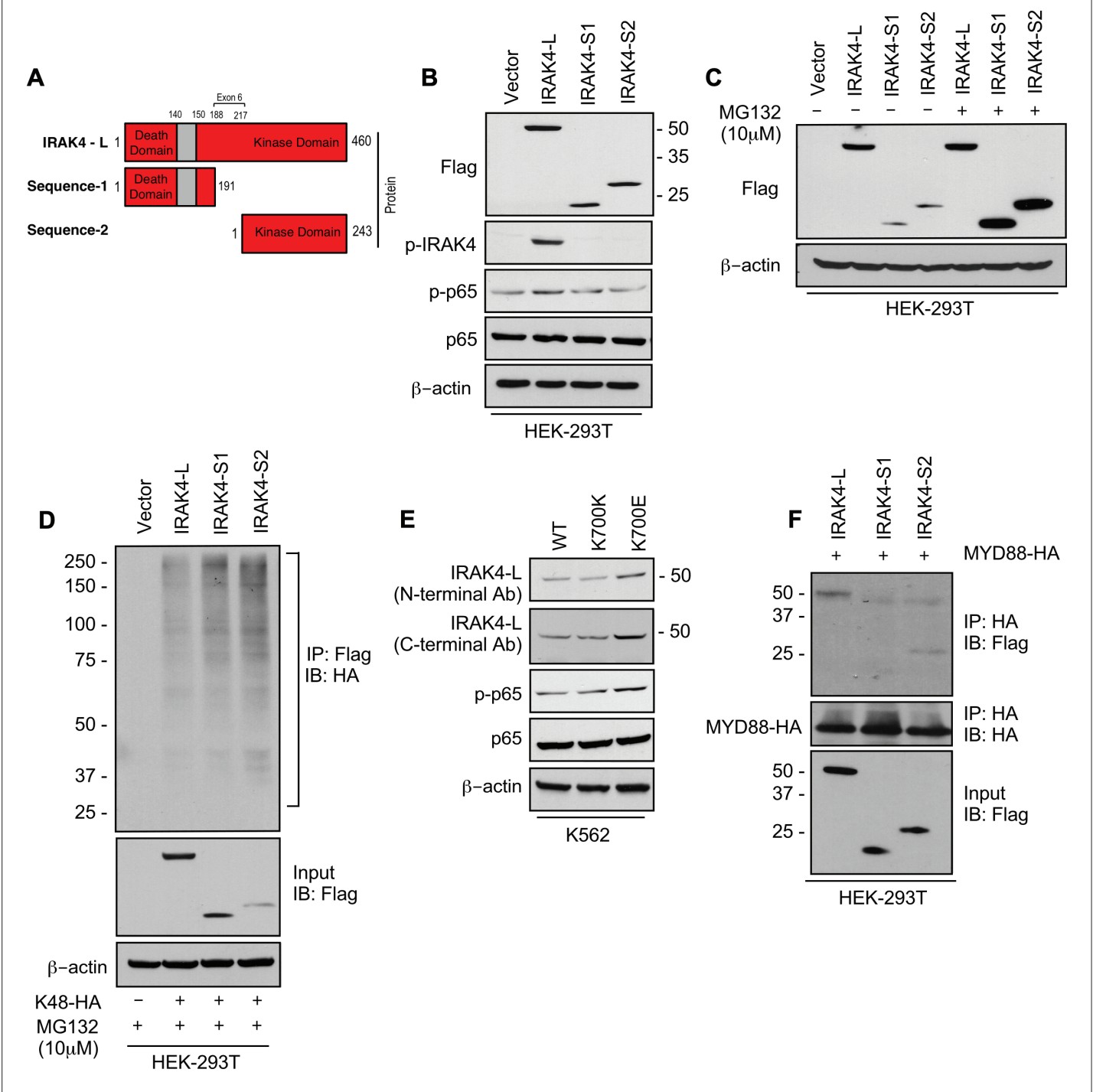

**Figure 2.** *SF3B1* mutation leads to longer, stable, and functionally active isoform of IRAK4. (**A**) Schematic showing that IRAK4 protein consists of a kinase domain and death domain that associates with MYD88. The segment of exon 6 that is not included in wild type (WT) controls encodes amino acids 188–217 and can lead to two smaller isoforms by generation of premature stop codon or use of alternative translational start site. (**B**) HEK-293T cells were transfected with Flag tagged plasmids of IRAK4-long isoform containing full length exon 6, IRAK4-S1, and IRAK4-S2. p-IRAK4, p-p-65, and

*Figure 2 continued on next page*

*Figure 2 continued*

p65 were determined by immunoblotting. (**C**) Flag tagged plasmids of IRAK4-long isoform containing full length exon 6, IRAK4-S1, and IRAK4-S2 were transfected in HEK293T cells and treated with proteasome inhibitor MG132 (10 uM) overnight followed by immunoblotting. IRAK4-S1 and IRAK4-S2 constructs that led to smaller IRAK4 protein bands had lower protein expression and were accumulated upon proteasomal inhibition. (**D**) Ubiquitination of ectopic IRAK4 was determined in HEK-293T cells transfected with indicated plasmids in presence of MG132 (10 μM) by immunoprecipitating with HA-specific antibody and immunoblotting with Flag. (**E**) Immunoblotting analysis for indicated proteins in isogenic K562 cells with *SF3B1* WT and K700E mutation. (**F**) HEK-293T cells were transfected with Flag tagged IRAK4 isoforms and HA tagged MYD88. MYD88 was immunoprecipitated with HA-specific antibody and its association with IRAK4 was probed by immunoblotting with specific antibodies.

The online version of this article includes the following source data and figure supplement(s) for figure 2:

**Source data 1.** Expression levels of IRAKa and its signaling partners in HEK293T cells after transfection with indicated plasmids.

**Source data 2.** Expression levels of IRAK4 isoforms transfected in HEK293T in presence of proteosome inhibitor.

**Source data 3.** Ubiquitination of ectopic IRAK4 in HEK293T in presence of MG132.

**Source data 4.** Expression levels of proteins in inflmmation pathway in K562 Ctl and K700E cells.

**Source data 5.** Association of IRAK4 isoforms with MyD88.

**Figure supplement 1.** Amino acid sequences of IRAK4-long and the shorter IRAK4-Sequence 1 and IRAK4-Sequence 2 isoforms.

Furthermore, we used isogenic cell line containing *SF3B1* K700E mutant and K700K control and demonstrated that the mutant cells contained heavily K63 ubiquitin CDK2 (*Figure 3E*). Since K63 ubiquitination leads to increased protein stability and interactions, we tested whether inhibiting CDK2 would lead to functional effects in patient samples containing the SF3B1 mutation. Cases of MDS with *SF3B1* mutations are characterized by ineffective hematopoiesis and treatment with CDK2 inhibitors in vitro led to increased hematopoietic colony formation and increased myeloid differentiation as evident by CD14 expression seen in three primary samples. (*Figure 3F*).

## IRAK4 inhibition leads to increased differentiation in *SF3B1* mutant MDS

The hallmark of MDS and AML is a block in differentiation leading to dysplastic maturation and cytopenia. We wanted to evaluate the efficacy of IRAK4 inhibition in *SF3B1* mutated MDS with a clinically relevant inhibitor, CA4948 (Emavusertib), which binds to the kinase domain on the protein (*Figure 4A*). CA4948 treatment is able to inhibit downstream NF-kB activation upon TLR engagement in leukemic THP1 cells (*Figure 4B*). This IRAK4 inhibitor also led to reduction in cytokine production in leukemic cells after engagements of Myd88/TLR pathways (*Figure 4C*). Bone marrow mononuclear cells from MDS patients with *SF3B1* mutation were cultured in the presence and absence of the IRAK4 inhibitor in vitro and then colonies were assessed by flow cytometry for myeloid differentiation. We observed a significant increase in myeloid colonies as well as increase in myeloid differentiation markers after IRAK4 inhibition in three distinct samples (*Figure 4D*). Treatment of healthy human CD34 stem and progenitor cells with IRAK4 inhibitor did not lead to any significant changes in colony numbers (*Figure 4E*). We also performed *IRAK4* knockdown with siRNAs in a primary MDS sample with *SF3B1* mutations and observed an increase in colonies and myeloid differentiation upon knockdown, validating IRAK4 as a target for relieving the differentiation block seen in MDS (*Figure 4F*).

## IRAK4 overexpression in MDS is associated with worse prognostic features and its inhibition leads to reduction in MDS/AML clones in vivo

Next we tested whether overexpression of IRAK4-long was associated with adverse clinical features in a large cohort of MDS samples (N=183) (*Pellagatti et al., 2010*). IRAK4-long expression was determined from an existing expression dataset from selected CD34+ cells and used to divide

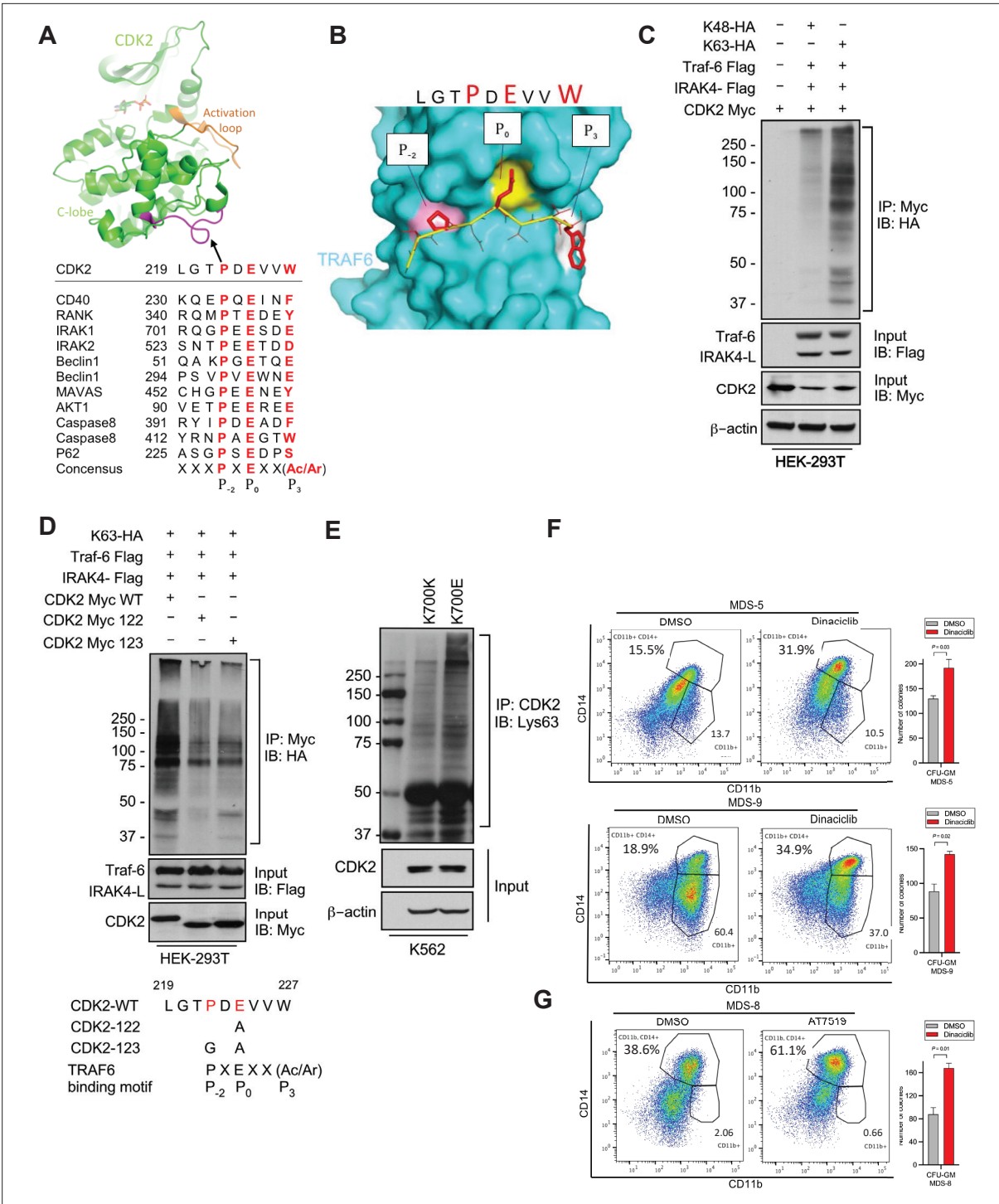

**Figure 3.** IRAK4/TRAF6 signaling regulates CDK2 ubiquitination in *SF3B1* mutant cells. (**A**) CDK2 structure (PDB:1GY3) bound to ATP is shown in cartoon representation. The putative TRAF6 binding peptide (magenta) is located at a loop between αG and αH in the C-lobe of the kinase. Sequence alignment of CDK2 peptide with common binding sequences with TRAF6. The consensus pattern with conserved P-2, P0, and P3 positions is shown at the bottom. (**B**) MM-GBSA docking of CDK2 peptide to substrate binding domain of TRAF6 (PDB:1LB5). The top 1 populated cluster is shown. TRAF6

*Figure 3 continued on next page*

*Figure 3 continued*

and CDK2 peptide are shown in ribbon and surface representation, respectively. Positions P-2, P0, and P3 correspond to three distinct sub-pockets in the binding site, formed largely by residues Y473, M450, G472 (pink), A458, G470, K469 (yellow), and H376, V374, R392 (white). Capital lettering of conserved residues P, E, and W within the shown peptides depicts successful docking in corresponding sub-pockets. (**C**) Lys63-linked proteins were immunoprecipitated in K562 cells with wild type (WT) and K700E SF3B1 mutation and CDK2 ubiquitination was determined by CDK2 specific antibody. (**D**) Ubiquitination of CDK2 was determined in HEK-293T cells transfected with indicated plasmids by immunoprecipitating with HA specific antibody and immunoblotting with either Lys48 or Lys63 antibody. (**E**) Lys63-linked proteins were immunoprecipitated in HEK-293T cells transfected with indicated plasmids and CDK2 ubiquitination was determined by Myc-tag specific antibody. Sequence alignment of optimum amino-acids sequence predicted for TRAF6 with WT and mutant CDK2 (CDK2-122 and CDK2-123) (**F–G**) Myelodysplastic syndromes (MDS) patient derived samples (bone marrow/peripheral blood) were treated with DMSO or Dinaciclib (10 nM) or AT7519 (10 nM) for 14 days on methylcellulose clonogenic assays. The samples were evaluated for colony formation CFU-GM (colony forming unit - granulocyte macrophage) and for myeloid differentiation on colonies and were subjected to flow cytometry analysis.

The online version of this article includes the following source data and figure supplement(s) for figure 3:

**Source data 1.** Ubiquitination of CDK2 in HEK293T cells.

**Source data 2.** Ubiquitination of CDK2 mutants in HEK293T cells.

**Source data 3.** Ubiquitination of CDK2 in K562 CTL and K700E cells.

**Figure supplement 1.** Modeling interaction between TRAF6 and CDK2 and CDK2 ubiquitination after treatment with IRAK4 inhibitors.

**Figure supplement 1—source data 1.** Ubiquitination of CDK2 in MDS-L cells with IRAK4 inhibitors CA-4948 (Emavusertib) and PF06650833.

the cohort into high and low IRAK4 based on median expression. We observed that cohort with higher IRAK4-long expression had lower platelet counts, higher RBC transfusion needs and higher leukemic blast counts, all indicative of worse clinical features (*Figure 5A*). Next, we established xenografts with MDS samples with SF3B1 mutation and treated the NSG mice with either IRAK4 inhibitor (CA4948) and placebo controls. Treatment with IRAK4 inhibitor led to decrease in MDS cells after 3–4 weeks of treatment in five distinct samples (*Figure 5B–D*, *Figure 5—figure supplement 1*, *Supplementary file 1*).

Since innate immune signaling pathways have been shown to play roles in leukemic stem cell propagation (*Barreyro et al., 2018*), we next evaluated IRAK4-long expression in a dataset derived from sorted AML/MDS HSCs and HC (*Barreyro et al., 2012*). IRAK4-long was found to be elevated in selected cases of AML HSCs, most significantly in adverse risk patients with complex cytogenetics (*Figure 5E*). Consistent with this finding, we derived a transcriptomic signature of IRAK4 overexpression from 183 MDS samples (*Figure 5—figure supplement 2*) and compared it to a published leukemic stem cell signature (*Eppert et al., 2011*) and found significant concordance (*Figure 5F*). Lastly, we tested the efficacy of IRAK4 inhibition in disease initiation in vivo. IRAK4 inhibitor and placebo treated xenograft mice were sacrificed and cells used for secondary xenografts. IRAK4 inhibition led to reduced MDS clones in secondary transplants, demonstrating decrease in disease initiating activity (*Figure 5G and H*, *Figure 5—figure supplement 3*, summarized in *Figure 5—figure supplement 4*).

## Discussion

MDS and AML are frequently incurable hematologic disorders characterized by cytopenia that are a major cause of morbidity and mortality. Mutations in splicing genes are one of the most common alterations in MDS and AML and predominantly affect *SF3B1, U2AF1, and SRSF2* (1). *SF3B1* is the most commonly mutated splicing gene in MDS and can also be mutated in solid tumors such as breast cancer, melanoma, and others (*Papaemmanuil et al., 2013*). Even though mutations in *SF3B1* can affect splicing of numerous genes, the exact pathways that are disrupted by aberrant splicing and lead to ineffective hematopoiesis are not well elucidated. We recently demonstrated that mutant *U2AF1 (S34F)* directly regulates IRAK4 exon 4 retention in MDS/AML, which results in expression of the longer IRAK4 isoform (IRAK4-L) (*Smith et al., 2019*). In the present report, we demonstrate that

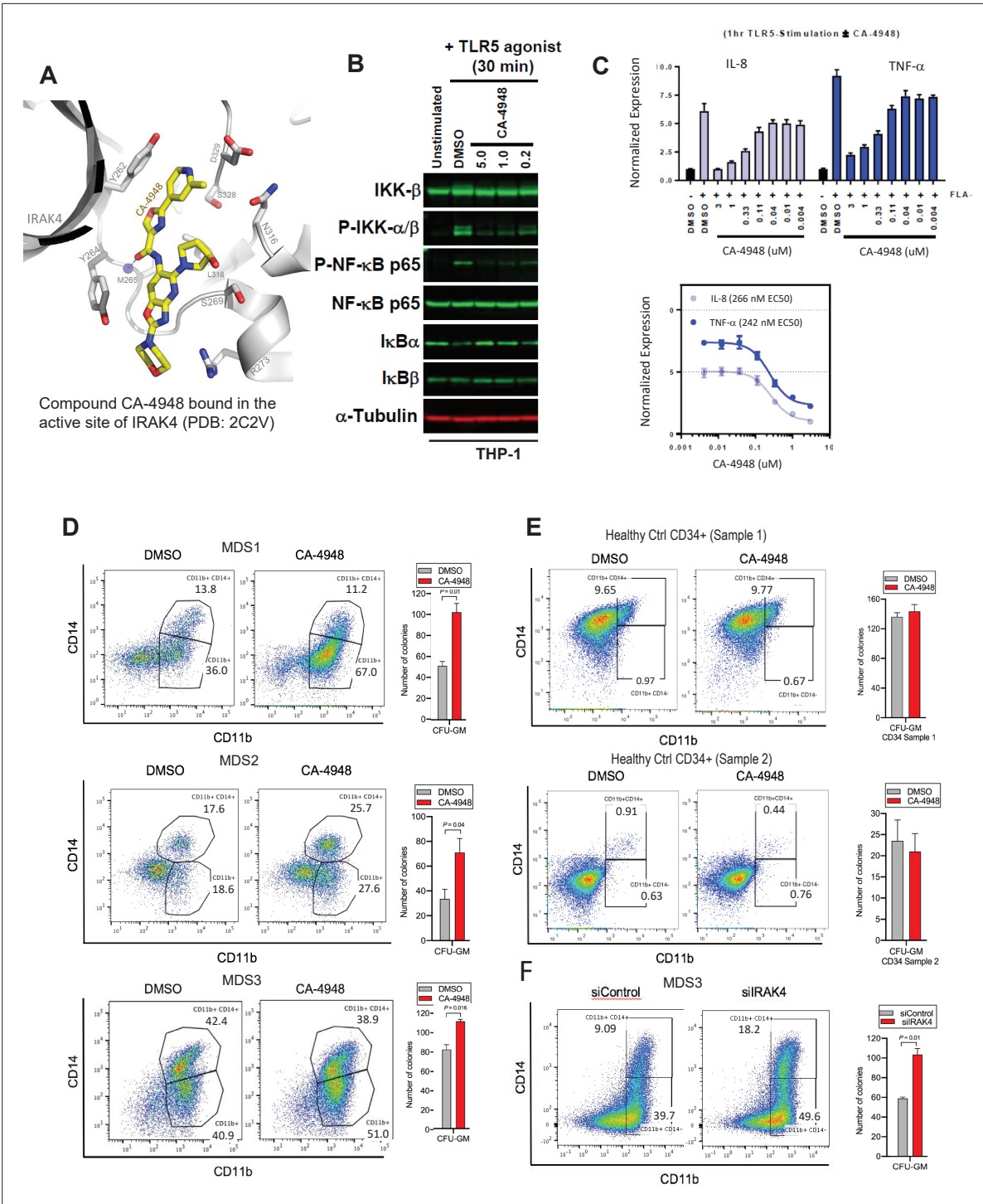

**Figure 4.** IRAK4 inhibition with CA-4948 promotes differentiation in myelodysplastic syndromes (MDS)/acute myeloid leukemia (AML). (**A**) Crystal structure of CA-4948 bound in the active site of IRAK4 (PDB:7C2V). Interacting residues of IRAK4 are shown in sticks. (**B**) Immunoblotting analysis for indicated proteins in THP-1 cells that were stimulated by toll like receptors (TLR)-5 and treated either with DMSO or CA-4948. (**C**) mRNA levels of IL8 and TNF-a in OCI-AML2 cells stimulated by TLR-5 ligand and were either treated with DMSO or CA-4948. (**D**) MDS patient derived samples with SF3B1

*Figure 4 continued on next page*

*Figure 4 continued*

mutation were treated with IRAK4 inhibitor (CA4948) or control in methylcellulose clonogenic assays and the analyzed for myeloid colony formation. Colonies were picked and analyzed by FACS for myeloid differentiation markers CD11b and CD14. (**E**) Healthy CD34+ stem and progenitor cells were grown in clonogenic assays with and without IRAK4 inhibitor (CA4948). Myeloid colonies were counted at day 14 and colonies were analyzed by FACS. (**F**) MDS patient sample with SF3B1 mutation was treated with siRNAs against IRAK4 and control and grown in clonogenic assays. The sample was evaluated for myeloid colony formation and for differentiation by analyzing their CD11b and CD14 expression with flow cytometry.

*SF3B1* mutations also lead to production of active IRAK4-L isoforms but by aberrant retention of exon 6 (*Figure 5—figure supplement 4*). The functional activation of the IRAK4 pathway by two different exon retention events by *U2AF1* mutations and *SF3B1* mutations suggest that this is an important functional event in MDS pathobiology.

While IRAK4 and its downstream pathways have been studied in innate immunity and in the context of inflammatory diseases, its molecular and cellular effects in MDS/AML are relatively understudied. Recent work has shown that innate immune signaling mediators such as IL1RAP, IL8/CXCR2, TLRs as well as myeloid derived suppressor cells are activated in MDS (*Barreyro et al., 2012*; *Dimicoli et al., 2013*; *Wei et al., 2013*; *Chen et al., 2013*; *Rhyasen et al., 2013*). Our findings further support the role of activated innate immune signaling pathways in MDS and suggest that interaction of IRAK4 with upstream TLR/MyD88 pathways is critical in MDS pathogenesis. Demonstration of overexpression of active IRAK4 isoforms in splicing mutant MDS suggests that these cells are primed to respond to upstream TLR and MyD88 activation. In fact, since active IRAK4 isoforms are expressed in different spliceosome mutant subsets, they potentially represent a shared downstream functional pathway that regulates MDS/AML malignant cell survival.

CA-4948 is a specific inhibitor of IRAK4 that is now being tested in clinical trials in MDS/AML and lymphomas (ClinicalTrials.gov: NCT04278768; NCT03328078) and in preliminary findings has shown a tolerable safety profile. Early clinical data appears encouraging in spliceosome mutant patients, supporting the validity of IRAK4 as a therapeutic target in SF3B1 mutant MDS/AML (*Garcia-Manero et al., 2021*). CA4948 also shows inhibitory activity against FLT3 and CLKs, potentially another reason for the antimalignant activity observed in our preclinical studies. Our data suggests that inhibiting IRAK4 pharmacologically leads to shrinkage of the MDS/AML clones in vivo and this correlates with decreased blast counts seen with CA4948 monotherapy in relapsed/refractory MDS/AML in early results. MDS/AML are characterized by block in hematopoietic differentiation that leads to neutropenia and subsequent infections. Our data suggests that IRAK4 inhibition promotes myeloid differentiation in SF3B1 mutant cases. This affect can be therapeutically advantageous in clinical trials and supports further testing.

## Acknowledgements

This work was supported by Cincinnati Children's Hospital Research Foundation, Leukemia Lymphoma Society, and National Institute of Health (R35HL135787, RO1HL111103, RO1DK102759, RO1HL114582), Gabrielle's Angel Foundation for Cancer Research, and Edward P Evans Foundation grants to DTS. AV is supported by Edward P Evans Foundation, National Institute of Health (R01HL150832, R01HL139487, R01CA275007), Leukemia and Lymphoma Society, Curis and a gift from the Jane and Myles P Dempsey family. AP and JB are supported by Blood Cancer UK (grants 13,042 and 19004). GC is supported by a training grant from NYSTEM. We acknowledge support of this research from The Einstein Training Program in Stem Cell Research from the Empire State Stem Cell Fund through New York State Department of Health Contract C34874GG. MS is supported by a National Institute of Health Research Training and Career Development Grant (F31HL132420).

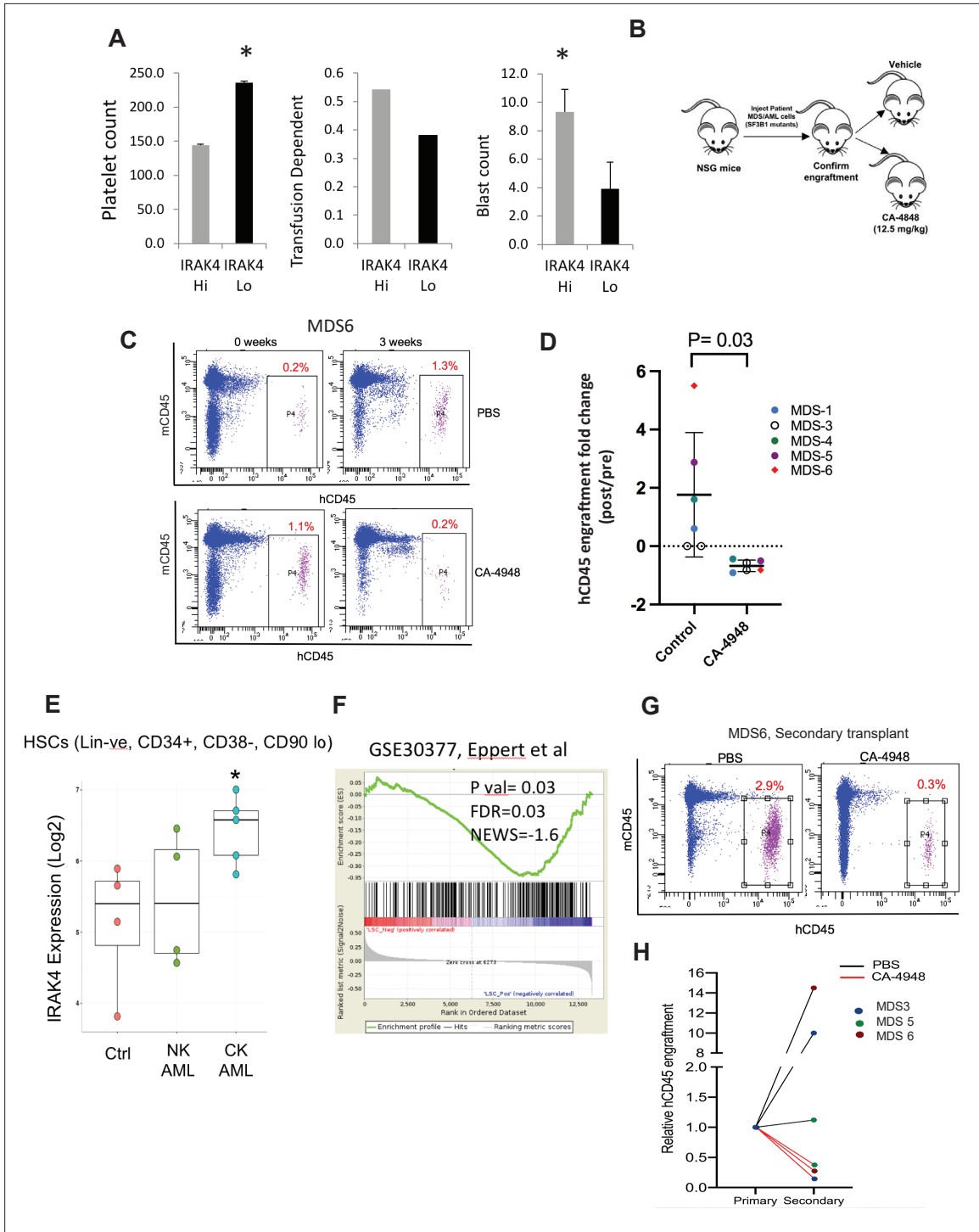

**Figure 5.** IRAK4 overexpression is associated with adverse clinical features in human myelodysplastic syndromes (MDS), and its inhibition leads to reduction in MDS clones in vivo. (**A**) IRAK4 expression in 183 samples from MDS patient bone marrow CD34+ cells was correlated with platelet counts, RBC transfusion dependency, and blast counts. Cohort with high *IRAK4* expression (>median) were associated with lower platelet counts, higher transfusion dependence, and higher blast counts (t-test, p<0.05). (**B**) NSG mice were xenografted with MDS cells with *SF3B1* mutation. After

*Figure 5 continued on next page*

*Figure 5 continued*

engraftment, mice were treated with either vehicle or IRAK4 inhibitor (CA-4849, 12.5 mg/kg) and bone marrow aspirates were used to evaluate human cells engraftment by flow cytometry. (**C–D**) Summary of human cell engraftment for xenografted mice treated with either IRAK4 inhibitor (CA-4948) or vehicle. Representative MDS sample xenograft with *SF3B1 K700E* mutation is shown. (**E**) IRAK4 expression in sorted HSCs (Lineage –ve, CD34+, CD38–) cells from MDS/acute myeloid leukemia (AML) cases (N=9) and controls (N=5). Significantly increased *IRAK4* expression is seen in samples with complex karyotype (CK) when compared to HC (t-test, p<0.05). (**F**) Gene expression signature associated with high *IRAK4* expression (>median, from set of 183 MDS CD34+ cells) was compared to know leukemic stem cell signature and showed significant similarity by GSEA analysis. (**G–H**) MDS bone marrow cells were isolated and purified from mice either treated with vehicle or CA-4948 in and then xenografted in secondary recipient NSG mice. Human cell engraftment was evaluated after 4 weeks by flow cytometry on bone marrow aspirates.

The online version of this article includes the following figure supplement(s) for figure 5:

**Figure supplement 1.** IRAK4 inhibition leads to reduction in myelodysplastic syndromes (MDS) clones in vivo.

**Figure supplement 2.** Gene expression signature associated with high IRAK4 expression.

**Figure supplement 3.** IRAK4 inhibition leads to reduction in myelodysplastic syndromes (MDS) clones in secondary transplant recipient mice.

**Figure supplement 4.** Proposed schematic of myelodysplastic syndromes (MDS) pathogenesis due to mis-splicing of IRAK4 because of SF3B1 mutations.

**Figure supplement 5.** STROBE flowchart showing study schema.

# Additional information

### Competing interests

Molly A Smith: MS owns stock options in Poseida Therapeutics. The author has no other competing interests to declare. Robert N Booher: RB is a former employee of Curis, Inc, and holds patents patents related to CA-4948 (WO-2019089580-A1). RB also owns stocks in Curis, Inc. The author has no other competing interests to declare. Murali Ramachandra: MR is an employee of Aurigene Inc. Maria Samson: MS owns stock options in Curis, Inc as an employee. The author has no other competing interests to declare. Robert K Bradley: RKB is a named inventor on patent applications related to treating cancers with SF3B1 mutations filed by Fred Hutchinson Cancer Research Center (METHODS AND COMPOSITIONS COMPRISING BRD9 ACTIVATING THERAPIES FOR TREATING CANCERS AND RELATED DISORDERS - PCT/US2020/039645; SYNTHETIC INTRONS FOR TARGETED GENE EXPRESSION - PCT/US21/56273).The author has no other competing interests to declare. Robert E Martell: REM is an employee of Curis, Inc and has received honoria payments and also holds stocks in the company. The author has no other competing interests to declare. Ulrich G Steidl: US held grants from Bayer Healthcare and Aileron Therapauetics, and received consultancy fees from Aileron Therapeutics, Stelexis Therapeutics, Pieris Pharmaceuticals and Trillium Therapeutics. US is Director at Stelexis Therapeutics and holds stocks in the company. The author has no other competing interests to declare. Daniel T Starczynowski: DS received consultancy fees from Kurome Therapeutics, Treeline Biosciences, Tolero Therapeutics, and Captor Therapeutics. DS also owns stocks in Kurome Therapeutics. The author has no other competing interests to declare. Amit Verma: AV has received research funding from GlaxoSmithKline, BMS, Jannsen, Incyte, MedPacto, Celgene, Novartis, Curis, Prelude and Eli Lilly and Company, has received compensation as a scientific advisor to Novartis, Stelexis Therapeutics, Acceleron Pharma, and Celgene, and has equity ownership in Throws Exception and Stelexis Therapeutics. The other authors declare that no competing interests exist.

### Funding

| Funder | Grant reference number | Author |
|---|---|---|
| National Cancer Institute | R01CA275007 | Amit Verma |
| Edward P. Evans Foundation | | Daniel T Starczynowski Amit Verma |
| Leukemia and Lymphoma Society | TRP | Daniel T Starczynowski Amit Verma |
| National Heart, Lung, and Blood Institute | R01HL150832 | Daniel T Starczynowski Amit Verma |

| Funder | Grant reference number | Author |
|--------|------------------------|--------|
| National Heart, Lung, and Blood Institute | R01HL139487 | Daniel T Starczynowski<br>Amit Verma |
| National Heart, Lung, and Blood Institute | RO1HL111103 | Daniel T Starczynowski<br>Amit Verma |
| National Heart, Lung, and Blood Institute | R35HL135787 | Daniel T Starczynowski<br>Amit Verma |
| National Institute of Diabetes and Digestive and Kidney Diseases | RO1DK102759 | Daniel T Starczynowski |
| Blood Cancer UK | 13042 | Andrea Pellagatti<br>Jacqueline Boultwood |
| Blood Cancer UK | 19004 | Andrea Pellagatti<br>Jacqueline Boultwood |

The funders had no role in study design, data collection and interpretation, or the decision to submit the work for publication.

## Author contributions
Gaurav S Choudhary, Conceptualization, Formal analysis, Supervision, Validation, Investigation, Visualization, Methodology, Writing – original draft, Writing – review and editing; Andrea Pellagatti, Conceptualization, Data curation, Formal analysis, Supervision, Validation, Investigation, Writing – original draft, Writing – review and editing; Bogos Agianian, Molly A Smith, Shanisha Gordon-Mitchell, Srabani Sahu, Sanjay Pandey, Nishi Shah, Srinivas Aluri, Ritesh Aggarwal, Sarah Aminov, Leya Schwartz, Violetta Steeples, Maria Samson, Milagros Carbajal, Kith Pradhan, Investigation; Tushar D Bhagat, Conceptualization, Investigation; Robert N Booher, Teresa V Bowman, Britta Will, Amittha Wickrema, Aditi Shastri, Robert K Bradley, Robert E Martell, Ulrich G Steidl, Supervision; Murali Ramachandra, Resources; Manoj M Pillai, Resources, Supervision; Evripidis Gavathiotis, Formal analysis, Supervision; Jacqueline Boultwood, Supervision, Project administration, Writing – review and editing; Daniel T Starczynowski, Supervision, Writing – review and editing; Amit Verma, Conceptualization, Resources, Data curation, Software, Formal analysis, Supervision, Funding acquisition, Validation, Investigation, Visualization, Methodology, Writing – original draft, Project administration, Writing – review and editing

## Author ORCIDs
Gaurav S Choudhary ⓘ http://orcid.org/0000-0001-5365-6706
Tushar D Bhagat ⓘ http://orcid.org/0000-0002-4527-5505
Sanjay Pandey ⓘ http://orcid.org/0000-0003-2948-1816
Robert K Bradley ⓘ http://orcid.org/0000-0002-8046-1063
Evripidis Gavathiotis ⓘ http://orcid.org/0000-0001-6319-8331
Jacqueline Boultwood ⓘ http://orcid.org/0000-0002-4330-2928
Amit Verma ⓘ http://orcid.org/0000-0002-5408-1673

## Ethics
Human subjects: The human samples used in this study were obtained with written informed consent approved by the IRB of the Albert Einstein College of Medicine.
This study was performed in strict accordance with the recommendations in the Guide for the Care and Use of Laboratory Animals of the National Institutes of Health. All of the animals were handled according to approved institutional animal care and use committee (IACUC) protocols (001371) of the Albert Einstein College of Medicine.

## Decision letter and Author response
Decision letter https://doi.org/10.7554/eLife.78136.sa1
Author response https://doi.org/10.7554/eLife.78136.sa2

## Additional files

### Supplementary files

• Supplementary file 1. Patient samples.

• Transparent reporting form

### Data availability

Publicly available dataset was used (https://www.ncbi.nlm.nih.gov/geo/query/acc.cgi?acc=GSE114922).

The following previously published dataset was used:

| Author(s) | Year | Dataset title | Dataset URL | Database and Identifier |
|---|---|---|---|---|
| Pellagatti A, Armstrong RN, Steeples V, Sharma E, Repapi E, Singh S, Sanchi A, Radujkovic A, Horn P, Dolatshad H, Roy S, Broxholme J, Lockstone H, Taylor S, Giagounidis A, Vyas P, Schuh A, Hamblin A, Papaemmanuil E, Killick S, Malcovati L, Hennrich ML, Gavin AC, Luft T, Hellström-Lindberg E, Cazzola M, Smith CWJ, Smith S, Boultwood J, Ad HO | 2018 | Impact of spliceosome mutations on RNA splicing in myelodysplasia: dysregulated genes/pathways and clinical associations | https://www.ncbi.nlm.nih.gov/geo/query/acc.cgi?acc=GSE114922 | NCBI Gene Expression Omnibus, GSE114922 |

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
