## [Editor Report]

This is an outstanding manuscript that sought to evaluate a mutation commonly seen in AML and MDS in the splicesome SF3B1. The authors demonstrate that this mutation leads to a shift in the production of a long-form of IRAK4 (called IRAK4-L), which is part of inflammatory signaling in immune cells. They demonstrate that IRAK4-L stabilizes the cell cycle protein CDK2, and targeting IRAK4-L with an inhibitor can induce differentiation and slow clonal uptake in murine transplantation models.

---

## [Decision Letter]

**Decision letter after peer review:**

Thank you for submitting your article "Activation of targetable inflammatory immune signaling is seen in Myelodysplastic Syndromes with SF3B1 mutations" for consideration by *eLife*. Your article has been reviewed by 2 peer reviewers, one of whom is a member of our Board of Reviewing Editors, and the evaluation has been overseen by a Reviewing Editor and Mone Zaidi as the Senior Editor. The following individual involved in the review of your submission has agreed to reveal their identity: Samir Parekh (Reviewer #2).

Essential revisions:

1) Please clarify the CDK2 issues raised by both reviewers.

2) Please address in detail the minor comments brought up by both reviewers.

*Reviewer #1 (Recommendations for the authors):*

Minor weaknesses focus on how certain items in the data presented in the figures are glossed over and not explained. Overall, an outstanding job and it was a pleasure to read.

The CDK2 piece of the story felt a little like a blind offshoot in the manuscript. Is there any data that can be shared from the human specimens regarding CDK2 to help tie differential ubiquitination, etc. back to the SF3B1 mutation?

Figure 1D: There is a smaller band below the putative IRAK4 in the SF3B1 lane. Please explain what that is and if it is reproducible.

Figure 4B and C: There appears to be a slight discrepancy in the concentrations with B using much higher concentrations than C. As a result, there is little recovery in B to the DMSO level. The (p) recovery is there but is very hard to see; might be worth attempting to quantitate.

Sup Figure 3: Generally, this figure is poorly put together with some axis language cut off and mcd45 and mCD45 annotated differently for some reason. The biggest issue that I would like to see addressed, however, is why in many of the figures do you have much higher engraftment rates in the CA-4948 group at 0 weeks. If this was done on purpose, then the rationale should be spelled out.

*Reviewer #2 (Recommendations for the authors):*

1. Can the authors expand on the specificity of the IRAK4 inhibitor CA4948? Could its inhibition of FLT3 be useful in myeloid malignancy setting?

2. The authors convincingly show decreased TRAF6 mediated Lys63 linked CDK2-ubiquitination in CDK2 mutant cells. Does CDK2-Ub decrease following IRAK4 inhibition in THP1 or MDS cell cells.

---

## [Author Response]

Reviewer #1 (Recommendations for the authors):Minor weaknesses focus on how certain items in the data presented in the figures are glossed over and not explained. Overall, an outstanding job and it was a pleasure to read.The CDK2 piece of the story felt a little like a blind offshoot in the manuscript. Is there any data that can be shared from the human specimens regarding CDK2 to help tie differential ubiquitination, etc. back to the SF3B1 mutation?

The reviewer raises an interesting point. We investigated CDK2 after an unbiased discovery of motif targeted by TRAF6. It has been shown in previous papers that decrease in CDK2 protein levels are critical for myeloid differentiation to occur (Ying et al., Ubiquitin-dependent degradation of CDK2 drives the therapeutic differentiation of AML by targeting PRDX2, Blood (2018) 131 (24): 2698–2711). Since block in differentiation is the hallmark of MDS/AML, we investigated Lys63 Ub of CDK2 in MDS/AML and also showed the ability of CDK2 inhibitors in promoting myeloid differentiation. We hope that future studies will evaluate CDK2 Ub patterns in primary samples, especially in splicing mutant cases.

Figure 1D: There is a smaller band below the putative IRAK4 in the SF3B1 lane. Please explain what that is and if it is reproducible.

The smaller band was not reproducible nor found in the endogenous exon/intron cassette so likely a technical artifact of the splicing reporter system. A deep analysis of MDS patient samples by RNA-seq also did not reveal any newer shorter splice sites. It is possible in the reporter system, a minor/cryptic splice site, could results in a slightly shorter band when over expressing SF3B1.

Figure 4B and C: There appears to be a slight discrepancy in the concentrations with B using much higher concentrations than C. As a result, there is little recovery in B to the DMSO level. The (p) recovery is there but is very hard to see; might be worth attempting to quantitate.

The reviewer raises an interesting point. We had used lesser number of doses in the western blot (Figure 4B) to highlight the downstream inhibition of NF-κB. In Figure 4C, we used a larger dose range to illustrate the biological effects better. We have quantified the blots (Author response image 1) and show a clear dose response inhibition with CA4948 (In response to stimulation with TLR5 agonist).

**Author response image 1. sa2fig1:** 

Sup Figure 3: Generally, this figure is poorly put together with some axis language cut off and mcd45 and mCD45 annotated differently for some reason. The biggest issue that I would like to see addressed, however, is why in many of the figures do you have much higher engraftment rates in the CA-4948 group at 0 weeks. If this was done on purpose, then the rationale should be spelled out.

Based on the reviewer’s suggestion, we have corrected the figure legends for Supp Figure 3. Xenografts with MDS samples, especially with lower risk SF3B1 mutant samples are technically challenging and yield variable rates of engraftment. The decision to treat/control was not based on engraftment numbers. Larger set of pdx will be performed in future studies.

Reviewer #2 (Recommendations for the authors):1. Can the authors expand on the specificity of the IRAK4 inhibitor CA4948? Could its inhibition of FLT3 be useful in myeloid malignancy setting?

The reviewer raises a good point. We have now added language in the discussion discussing the potential impact of FLT3 inhibition with CA4048 in myeloid malignancies.

2. The authors convincingly show decreased TRAF6 mediated Lys63 linked CDK2-ubiquitination in CDK2 mutant cells. Does CDK2-Ub decrease following IRAK4 inhibition in THP1 or MDS cell cells.

Based on the reviewer’s excellent suggestion, we performed experiments evaluating this effect using two IRAK4 inhibitors in MDS-L cell line (Supp Figure 2B).

Immunoprecipitation with CDK2 and immunoblotting with anti-Lys63 ub antibody shows decreased overall smear after IRAK4 inhibition.